# Adaptive Reflection Detection and Control Strategy of Pointer Meters Based on YOLOv5s

**DOI:** 10.3390/s23052562

**Published:** 2023-02-25

**Authors:** Deyuan Liu, Changgen Deng, Haodong Zhang, Jinrong Li, Baojun Shi

**Affiliations:** 1State Key Laboratory of Reliability and Intelligence of Electrical Equipment, Hebei University of Technology, Tianjin 300401, China; 2Hebei Key Laboratory of Robot Sensing and Human-Robot Integration, Tianjin 300401, China; 3School of Mechanical Engineering, Hebei University of Technology, Tianjin 300401, China

**Keywords:** inspection robot, YOLOv5s, k-means clustering algorithm, reflection detection, pose transformation

## Abstract

Reflective phenomena often occur in the detecting process of pointer meters by inspection robots in complex environments, which can cause the failure of pointer meter readings. In this paper, an improved k-means clustering method for adaptive detection of pointer meter reflective areas and a robot pose control strategy to remove reflective areas are proposed based on deep learning. It mainly includes three steps: (1) YOLOv5s (You Only Look Once v5-small) deep learning network is used for real-time detection of pointer meters. The detected reflective pointer meters are preprocessed by using a perspective transformation. Then, the detection results and deep learning algorithm are combined with the perspective transformation. (2) Based on YUV (luminance-bandwidth-chrominance) color spatial information of collected pointer meter images, the fitting curve of the brightness component histogram and its peak and valley information is obtained. Then, the k-means algorithm is improved based on this information to adaptively determine its optimal clustering number and its initial clustering center. In addition, the reflection detection of pointer meter images is carried out based on the improved k-means clustering algorithm. (3) The robot pose control strategy, including its moving direction and distance, can be determined to eliminate the reflective areas. Finally, an inspection robot detection platform is built for experimental study on the performance of the proposed detection method. Experimental results show that the proposed method not only has good detection accuracy that achieves 0.809 but also has the shortest detection time, which is only 0.6392 s compared with other methods available in the literature. The main contribution of this paper is to provide a theoretical and technical reference to avoid circumferential reflection for inspection robots. It can adaptively and accurately detect reflective areas of pointer meters and can quickly remove them by controlling the movement of inspection robots. The proposed detection method has the potential application to realize real-time reflection detection and recognition of pointer meters for inspection robots in complex environments.

## 1. Introduction

At present, there are a large number of pointer meters in petrochemical scenes, most of which are manually inspected. Missing detection and false detection often occur, and there are great safety hazards during detection. With the development of robot technology in recent years, more and more inspection robots have been used in petrochemical scenes to replace manual inspection tasks. In petrochemical scenes, the working environment of inspection robots is very complex, including adverse factors such as reflection, occlusion, water mist, and smoke. Thus, it is very difficult to detect and identify pointer meters in such a working environment. Among these factors, a reflective phenomenon is inevitable in the process of pointer meter detection, which is one of the main factors affecting pointer meter readings [1]. Generally speaking, a small reflective area does not affect the meter’s reading, which can be processed by an image inpainting algorithm or uneven lighting correction algorithm [2]. However, the influence of a large reflective area is obvious. In this case, the pointer meter dial information may be seriously lost, and it is hard to obtain a high-quality image only by image processing algorithm. Therefore, it is necessary to be supplemented with hardware adjustment for image acquisition. It is particularly important to accurately detect the reflective area and then adjust it according to the detection results.

In order to solve the problem of reflection detection, Sato et al. [3] used the patio-temporal information of images to detect and remove reflective areas based on time-varying image sequences. Liu et al. [4] estimated the relative intensity of the sun and the sky by extracting the plane feature point group of each frame in the video stream. Although this kind of method has good processing results, it is not practical because it processes sequence images and needs large computing power. Zhang et al. [5] used the principle of polarized light to summarize the distribution of the degree of polarization of each pixel in images, and then separated the transmitted light and the reflected light. Islam et al. [6] used the Lytro Illum camera to capture MSPLF (multispectral polarimetric light field) imagery and then used the polarization information of the image to determine reflective areas. Hao et al. [7] designed a new multi-band polarization imager to collect images, which can acquire all the spectral and polarization information, and proposed a joint multi-band-polarization characteristic vector constraint to detect reflective areas. These methods require professional equipment, special types of images, and complex processing. Therefore, they are limited in practical application. Zhai et al. [8] used the color information of the image to extract the saliency region by extracting the chromaticity difference between each pixel and the neighborhood pixel. Guo et al. [9] estimated the pixel brightness by the maximum value information in the RGB channel of each pixel. The processing effect is good to use some attributes of each pixel information, but the related computation is large and complex. In addition, for a reflective area of pointer meters, the brightness component is relatively saturated, and the two-color reflection model is generally not satisfied in this area [10]. Therefore, the method of decomposing pixel features is constrained. Zhang et al. [11] used grayscale images for mean filtering and used the maximum between-class variance to determine the threshold of image binarization for reflective area segmentation. Cao et al. [12] determined the reflective area by solving the non-mirror reflectance estimation map and then solving the diffuse reflectance chromaticity estimation of each pixel. Kang et al. [13] converted images to the HSI model and filtered the reflective pixels by the combination of the k-means clustering algorithm and filter. However, these methods have limited applications due to their shortcomings in precision, speed, and complex illumination conditions. Asif et al. [14] used the intrinsic image layer separation (IILS) technique to detect reflective areas. However, this method is easy to misdetect the high saturation areas of the high saturation image as the reflective areas. Nie et al. [15] used color change and gradient information to determine reflective areas. This method is suitable for images with prominent color features. However, pointer meter images are not, so this method has limitations.

Aiming at the shortcomings of these methods available in the literature, an adaptive detection method of pointer meter reflective areas is proposed based on a deep learning algorithm. There are two main contributions to this paper. Firstly, we propose an improved k-means clustering algorithm that can accurately and quickly detect reflective areas. Secondly, based on the detection results, we propose a new pose control strategy for inspection robots to adjust reflective areas, which can increase their reading accuracy and efficiency. Compared with some mainstream reflection detection algorithms, the main advantages of our proposed method are strong adaptability, low dependence on equipment, and simple. The disadvantage of this method is that the detection accuracy of some images with large color saturation is slightly lower. It is because the YUV (luminance-bandwidth-chrominance) model is not very suitable for highly saturated color images.

The rest of this article includes five sections. A pointer meter detection approach based on deep learning is introduced in Section 2. An improved k-means algorithm based on curve fitting is described in Section 3. A robot pose control strategy is proposed in Section 4. In Section 5, an experimental platform is described first and followed by experimental results and discussions to verify the performance of the proposed method. The main conclusions are summarized in the last section.

## 2. Pointer Meter Detection Based on Deep Learning

### 2.1. Deep Learning Image Acquisition

With the development of deep learning, more and more models are applied to machine vision, such as GANs (Generative Adversarial Networks), VGG (Visual Geometry Group), U-Net, Faster RCNN (Faster Region CNN), YOLO (You Only Look Once), etc. They greatly improve the automation and intelligence of target inspection [16,17,18,19]. So, in this paper, a deep learning network is adopted to detect pointer meter panels. YOLO is one of the best deep learning networks in the field of target detection. It is fast and accurate to achieve real-time target detection. Therefore, it is widely used in all walks of life, especially in the field of robots. YOLO has developed from the original v1 to the current v8. Among them, the most widely used version is v5. Some scholars have proposed v6, v7, and the latest v8 version based on v5. However, this paper chooses YOLOv5 as the network framework due to the universality and stability of the current use, etc.

YOLOv5 has strong flexibility and fast recognition speed. It is easy to deploy on inspection robots and has good recognition accuracy. Therefore, the most lightweight YOLOv5s network is selected on the premise of ensuring recognition accuracy [20].

YOLOv5s network structure is shown in Figure 1. It consists of the input layer (input), the backbone network (backbone), the bottleneck layer network (neck), and the detection layer (output). The backbone and neck are composed of Focus, CBL (Convolutional, Batch normalization, Leaky Relu), CSP (Cross Stage Partial), and SPP (Space Pyramid Pooling) [21]. Images are input by Input layer, and their feature extractions are performed through backbone network and the neck network. Three feature maps of different sizes are obtained at output layer to detect targets of different sizes.

In this paper, 5102 pointer meter images are collected as dataset, and the training set and verification set are divided according to the ratio of 7:3. LabelImg software is used to label each meter image. Category contains only “pointer”. Pytorch is selected as the deep learning framework. The initial learning rate is 0.01, the termination rate is 0.2, and the number of training rounds (epoch) is 300. The training results of precision and recall rates, which are used as evaluation indicators, are shown in Figure 2.

The precision rate is relative to the prediction results, which indicates how many samples are among the positive samples predicted. The recall rate is relative to the initial samples, which indicates how many positive samples are correctly predicted in the samples.
(1)$$Precision=\frac{TP}{TP+FP}$$
(2)$$Recall=\frac{TP}{TP+FN}$$
where *TP* (true positive) represents the number of positive samples which are truly detected by the model, *FP* (false positive) represents the number of positive samples which are falsely detected by the model, while *FN* (false negative) represents the number of negative samples for model false detection.

Figure 2 shows that with the increase in training rounds, the precision and recall rates tend to be stable.

The detection results of some pointer meters are shown in Figure 3. If the confidence level can reach the set threshold for the probability of a pointer meter (the maximum is 1, which is set as 0.7 in this paper), the pointer meter will be selected in its image with a box and its confidence level. After testing, the pointer meter recognition rate in this paper is above 95%.

### 2.2. Image Perspective Transformation

Generally, when a pointer meter is identified by human eyes, if the line of sight is perpendicular to the pointer meter dial, the captured pointer meter image is a circle. However, when a robot is used to collect images, the camera plane is often not parallel to the pointer meter panel, so the collected pointer meter image may be in an ellipse. In this case, the pointer meter images to be inspected are distorted images [22]. Perspective transformation is to project an image onto a new visual plane to transform an ellipse into a circle. The circle is obtained using the condition that the perspective center, image point, and target point are collinear. Therefore, the perspective transformation principle is adopted to transform an ellipse into a circle to obtain the front view of a pointer meter image. The schematic diagram is shown in Figure 4.

Figure 4 shows that the perspective transformation is to transform an ellipse perspective into a large circle outside. The relationship between the corresponding points is:(3)$$p_{i}(x,y)\to {p_{i}}^{\prime }(x,y)$$
where $p_{i}(x,y)$ represents the point on the ellipse, and ${p_{i}}^{\prime }(x,y)$ represents the point on the circle obtained by perspective transformation.

The perspective transformation formula is expressed in Formula (4):(4)$$(x,y,w^{\prime })=(u,v,w)\left[\begin{array}{ccc} a_{11} & a_{12} & 0\\ a_{21} & a_{22} & 0\\ 0 & 0 & 1 \end{array}\right]$$
where $\left(x,y,w^{\prime }\right)$ represent the coordinates of the circle on the transformed image. (u,v,w) represent the coordinates of the ellipse on the original image. The matrix represents the transformation matrix between the original image and the transformed image. $w^{\prime }$ and w are both equal to 1.

According to the geometric relationship between the straight line, the ellipse, and the large circle, four pairs of the corresponding points can be found for perspective transformation. Formula (5) gives the corresponding geometric relationship:(5)$$\{\begin{array}{c} y=\pm \frac{a}{b}x\\ \frac{y^{2}}{a^{2}}+\frac{x^{2}}{b^{2}}=1 \end{array}(a>b>0)\;or\{\begin{array}{c} y=\pm \frac{a}{b}x\\ \frac{x^{2}}{a^{2}}+\frac{y^{2}}{b^{2}}=1 \end{array}(a>b>0)$$

In order to reduce the computational cost and speed up the calculation, deep learning is combined with an ellipse detection algorithm [23]. Firstly, a pointer meter is detected using YOLOv5s, and the detected pointer meter area is selected with a box. Then, the ellipse detection algorithm is applied only for the selected area, which reduces the detection range of irrelevant areas and greatly improves the detection speed. Figure 5a is the boxed area detected using YOLOv5s, and Figure 5b is the ellipse detection of the detection result.

After the ellipse is detected, the pointer meter perspective view angle is corrected by perspective transformation. The correction result of Figure 5a is shown in Figure 6. It can be seen that the pointer meter panel is changed from an inclined angle to a positive angle.

Once the reflection of a pointer meter is detected during its detection process, it is necessary to make some changes to eliminate the reflection. The changes are based on an improved k-means algorithm and a robot pose control strategy.

## 3. Improved K-Means Algorithm Based on Curve Fitting

In this section, an improved k-means algorithm based on curve fitting is described using the YUV color model.

### 3.1. Color Model

A color model is a set of visible light in a 3D color space. It contains all colors of a color domain. The same object constitutes different color models from different measuring procedures, such as RGB, HSV, LAB, and YUV. Because this article mainly needs the brightness information of images, it mainly needs a color model to separate brightness information from chromaticity information.

Figure 7 shows the commonly used image color models. RGB has no brightness channel. HSV and LAB are intuitive color models. They are commonly used in image editing tools and are not suitable for lighting models. YUV model takes the brightness information as a separate component, which is separated from the chromaticity information U and V. It is suitable for the lighting model analysis. Therefore, in this paper, YUV is selected as the color model for image processing [24].

Therefore, it is necessary to convert pointer meter images from the RGB model into the YUV model. In the YUV model, Y is the brightness component, and U and V are the chroma components. Formula (6) gives the relationship between the RGB model and the YUV model.
(6)$$\left\{ \begin{array}{l} Y=0.299R+0.587G+0.114B\\ U=-0.147R-0.289G+0.436B\\ V=0.615R-0.515G-0.100B \end{array}\right.$$

Image histogram can be used to show the brightness range and the number of pixels in an image. In this paper, image histogram statistics are made on Y in images converted to YUV color model. A pointer meter image and its brightness histogram are shown in Figure 8. The horizontal axis of Figure 8b represents the brightness range, which is set as [0, 255]. The larger the horizontal axis value is, the higher the pixel energy value is, and the brighter the image is. The vertical axis represents the corresponding number of pixels, and the red polyline represents the brightness pixel distribution of the pointer meter image.

### 3.2. Polynomial Curve Fitting

For a nonlinear curve of data in the brightness component histogram, the law cannot be obtained by direct processing. Thus, it is often fitted into a linear model by curve fitting. In various curve fitting methods, polynomial fitting has the characteristics of rapid modeling, a significant effect on small data, and simple relationships. In this paper, the polynomial fitting method selected for the data is small in size, simple in the relationship, and single in attribute, which fits the conditions and requirements of polynomial fitting [25].

The mathematical expression of the polynomial fitting is expressed in (7):(7)$$y(x,\omega )=\sum_{j=0}^{M}\omega_{j}x^{j}$$
where *M* is the highest order of the polynomial, $x^{j}$ representing the *j*-th power of *x*, and $\omega^{j}$ represents the coefficient of $x^{j}$.

Gradient descent method is used to solve the polynomial. The objective loss function is expressed in (8).
(8)$$L(\omega )=\sum_{i=1}^{N}{\left\{y(x_{i},\omega )-t_{i}\right\}}^{2}$$
where *N* represents the number of samples, $t_{i}$ corresponds to the true value. 

The polynomial order *M* determines the polynomial fitting effect. If the order is small, there will be under-fitting with large fitting curve deviation and serious feature loss. While if the order is large, there will be over-fitting with being sensitive to noise and poor generalization ability [26]. In order to determine the optimal order *M*, the evaluation index of reflective area marking accuracy (in Section 5.2) is used as the fitting accuracy. The relationship between the fitting accuracy and fitting times is shown in Figure 9.

Figure 9 shows that when the fitting times are less than 12, the fitting accuracy is not good, and there is an under-fitting phenomenon. When the fitting time is 12, the fitting accuracy is optimal. With the increase in the fitting times, the fitting accuracy fluctuates within a small range, indicating that the increase in the fitting times has no obvious effect on the fitting accuracy. When the fitting times are more than 18, the fitting accuracy shows a downward trend and over-fitting occurs.

In order to determine the final order *M*, the method proposed in [27] is adopted to analyze the complexity of the k-means algorithm from the perspective of energy, as shown in (9):(9)ΔQ=Tb(klnn−nlnk)
where *T* is the temperature, *b* is the Boltzmann constant, *n* is the number of samples, *k* is the number of clusters, *n* is much larger than *k* in this paper. ΔQ>0 represents the energy released by the system and ΔQ<0 represents the energy input by the system. For calculation, only ΔQ<0 can be satisfied, and the greater ΔQ is, the lower the complexity of the algorithm is. Therefore, when *T*, *b,* and *n* are all determined and *n* much larger than *k*, the smaller *k* is the better it is. In curve fitting, the fewer the fitting times, the peak number will decrease accordingly, and the smaller *k* it is. Therefore, under the premise of the best fitting accuracy, it is determined that it is the best when *M* is equal to 12. The comparison of under-fitting, fitting, and over-fitting curves are shown in Figure 10.

The black marks in Figure 10 represents the peak and valley points of the fitting curves. It appears that Figure 10a which is under-fitting curve has only a small number of marked points, especially in the range of [150, 250], missing a lot of key information. Figure 10c which is over-fitting curve has too many marked points, especially in the range of [0, 150], containing a lot of irrelevant information. Figure 10b shows that the nonlinear curve fitting of the original data is fitted into a smooth curve with peak and valley information. The peak points are defined as $C_{m}(m=1,2,3,\ldots )$, the valley points are defined as $T_{n}(n=1,2,3,\ldots )$. The peak and valley points are unified to the feature point set *P*:(10)$$P=\left\{C_{m},T_{n}\right\}$$

A peak is formed when the curve trend goes up and then goes down at the stationary point. Similarly, a valley is formed when it goes down first and then goes up at the stationary point. Therefore, no peak or valley will be formed at the beginning and end of the curve, that is, at x = 0 and x = 255. However, for data integrity, it is necessary to add these two points to the feature point set *P* to form a complete feature point set *Pa*:(11)$$P_{a}=\left\{0,C_{m},T_{n},255\right\}$$

### 3.3. Principle and Improvement of k-Means Algorithm

#### 3.3.1. Principle of k-Means Algorithm

The basic principle of the k-means algorithm is to cluster *n* objects $X_{i}=\{X_{1},X_{2},\ldots ,X_{n}\}$ with k center points $C_{j}=\{C_{1},C_{2},\ldots ,C_{k}\}$, and gather them into *k* clusters according to the similarity among the objects. Each object has only one corresponding cluster, and the value of each cluster center is continuously updated iteratively until the iteration conditions are met.

Among these objects, the similarity between objects is measured by calculating the distance, which is generally using Euclidean distance:(12)$$dis(X_{i},C_{j})=\sqrt{\sum_{f=1}^{\gamma }{\left(X_{if}-C_{jf}\right)}^{2}}$$
where γ represents the number of attributes of objects, $X_{if}$ represents the *f*-th attribute of the *i*-th object, $C_{jf}$ represents the *f*-th attribute of the *j*-th cluster. In this paper, there is only one attribute of brightness, therefore γ=1.

Although the k-means algorithm is simple and fast, the selection of the k value and its initial values has a great influence on the clustering effect [28]. The main disadvantages are shown in the following two aspects: (1)The *k* value is uncertain. The *k* value of the traditional k-means algorithm is given manually, and the *k* value for different objects is also different. Selecting the *k* value improperly will affect the clustering effect.(2)The clustering effect is sensitive to initial clustering center values. The initial values are generally given randomly, and they have relatively large contingency and are easy to fall into local convergence. Therefore, it cannot achieve the goal of global convergence.

#### 3.3.2. Improved k-Means Algorithm

According to the two shortcomings of traditional k-means clustering algorithm, following improvements are made in this paper:

(1)According to the peak information of the fitting curve to determine the *k* value adaptively.

A peak point represents that the number of pixels corresponding to the brightness value is the largest in the range between the two adjacent valleys on the left and right sides of the point. This point can represent the brightness information within the threshold of the two adjacent valleys. Additionally, in terms of one peak, the range from its right side of the peak to the adjacent valley can be considered “bright”, while the range from its left side to the adjacent valley can be considered “dark”. Thus, each peak represents two brightness levels. Therefore, when the idea is extended to the overall situation, the *k* value corresponds to twice the number of peaks *m*, so it is determined according to the peak points $C_{m}(m=1,2,3,\ldots )$ mentioned in Section 2.2, i.e., k=2×m. It is worth noting that the brightness level represented by each peak here is local. Some small peaks will be swallowed up by large peaks during clustering, and then the brightness levels will be re-divided. Therefore, the global brightness levels can be divided only after the clustering is completed.

(2)According to the peaks and valleys information of the fitting curve to determine the values of the initial cluster centers.

According to the conclusion of improvement (1), each peak $C_{m}(m=1,2,3,\ldots )$ corresponds to two brightness levels, which corresponds to two initial clustering centers. In order to avoid the phenomenon of the local optimal solution, in this paper, the set of points $P_{a}$ is used as the reference to select the initial cluster values, i.e., in $P_{a}$, one initial value $I_{cen}$ of a cluster center is randomly selected from each adjacent two points.
(13)$$I_{cen,t}=P_{a}{}_{{}_{t}}+\alpha (P_{a_{t+1}}-P_{a_{t}}),(0<\alpha <1,\{t|0\leq t<m+n+2,t\in N\})$$
where $I_{cen,t}$ is the initial value of the cluster center, α is a random number of 0~1, *t* is the number of points in *P_a_*. By adopting this value-taking method, it can not only ensure the randomness of the values but also cover the overall data to avoid falling into the situation of the local optimal solution, which is more reasonable.

After the *k* value and the initial clustering center values are determined, the brightness values $\Omega_{i}(i=1,2,\ldots ,k)$ are obtained by k-means clustering. At the same time, the global brightness level number is set as *k*. The larger *i* corresponds to the higher brightness level which is brighter reflecting on the image. In addition, the reflective area generally corresponds to the area with the highest brightness level $\left\{x|\Omega_{k}\leq x\leq 255,x\in N\right\}$.

In this paper, the k-means algorithm is combined with polynomial curve fitting, curve peak, and valley information. A new image processing algorithm is developed to determine the optimal number of clusters and the initial cluster centers. The steps of the improved k-means algorithm are as follows:(1)First, an image is converted to a YUV color model to calculate the image brightness information histogram;(2)The brightness information histogram is fitted into a smooth curve by a 12th fitting curve;(3)Count the peak and valley information and add the two end-points to form the feature point set $P_{a}$;(4)Determine the optimal number of clusters k=2×m according to the number of peaks *m*;(5)Determine the initial clustering center $I_{cen}$ values according to the feature set of points $P_{a}$;(6)Finally, *k* clusters, are obtained by clustering calculation, i.e., there are *k* brightness levels. Eventually, the reflective area can be determined based on the highest brightness level.

The improved k-means algorithm is used to determine the robot pose control strategy.

## 4. Inspection Robot Pose Adjustment

If there is a large area of reflection on a pointer meter, it is hard to obtain enough image information by image processing only, which affects the accuracy of reading and even cannot read at all. Therefore, it is necessary to adjust the camera shooting angle with inspection robot, and remove the reflection detected based on light reflection law [29]. The reflection law is shown in Figure 11.

For a reflective pointer meter, after its image perspective transformation, the improved k-means algorithm will be used to detect its reflective areas. Then, the specific position information of the reflective areas on the pointer meter can be obtained to determine the motion direction and distance for the robot’s adjustment.

### 4.1. Determination of Motion Direction

In this paper, to prevent losing the pointer meter image due to the robot’s movement, the center c of the detection target with radius r is designed as the center of the moving path, which is a circular arc with radius R of the projection distance between the detection target and the camera [30]. Figure 12 shows the motion diagram of robot pose adjustment.

As shown in Figure 12, two arrow arcs in different directions represent the two path adjusting possibilities of the robot. The moving direction of the robot is determined by the central axis of the pointer meter [31] and the reflective areas on it. Firstly, the pointer meter contour circle is divided into two semicircles through the central axis, and then the reflective detection results are mapped on the two semicircles for comparison. The robot will move in the direction of larger reflective area on the two sides of the pointer meter. As shown in Figure 13, the red dotted line is the central axis of the pointer meter.

### 4.2. Determination of Moving Distance

After the moving direction is determined, in order to give the robot an exact moving distance, it is calculated according to the geometric relationship between the robot position, pointer meter position, incident light angle, and reflected light angle. In this paper, the reflective area on the right side of the pointer meter axis is taken as an example, and the robot position relationship is shown in Figure 14.

In Figure 14, the black thick line Me represents the pointer meter dial (radius *r*), and the Cartesian coordinate system xoy is established based on the center of the pointer meter. The red circle arc C with two arrows is the motion path of the robot. *Ak* and *kb* are the incident light and reflected light of the farthest reflection point *k* (shown in Figure 15 for details) and the corresponding position of the robot is *b*. *a’k’* and *k’b’* are the expected incident light and reflected light and the corresponding robot correction position is *b’. kk’* represents the maximum reflection length and *d* is the distance from the farthest reflection point to the central axis.

According to the reflective area diagram shown in Figure 15, the position of point k in Figure 14 can be obtained. Then, the position of point *b’* can be obtained according to the geometric relationship shown in Figure 14. Thus, the arc length corresponding to *bb’* in Figure 14 is the trajectory of the robot to move. Formula (14) represents the geometric relationship between the robot and the reflective point:(14)$$\{\begin{array}{c} x^{2}+y^{2}-R^{2}=0\\ Rx-dy-Rr=0 \end{array}$$
where (x,y) is the coordinate position of the robot, *R* is the projection distance of the camera relative to the pointer meter, *r* is the pointer meter radius, and the straight line *k’b’* is expressed as Rx−dy−Rr=0.

Substituting the coordinate position $\left(x_{b^{\prime }},y_{b^{\prime }}\right)$ of point *b’* into Formula (14) yields:(15)$$\{\begin{array}{c} x_{b^{\prime }}=\frac{R^{2}r-Rd\sqrt{R^{2}+d^{2}-r^{2}}}{R^{2}+d^{2}}\\ y_{b^{\prime }}=\frac{Rdr+R^{2}\sqrt{R^{2}+d^{2}-r^{2}}}{R^{2}+d^{2}} \end{array}$$

According to the relationship between the arc length and its angle, the arc length $l_{bb^{\prime }}$ corresponding to *bb’,* which is the actual motion distance of the robot, can be obtained.
(16)$$l_{bb^{\prime }}=\frac{\pi }{180}\arctan\left|\frac{x_{b^{\prime }}}{y_{b^{\prime }}}\right|$$

By determining the moving direction and moving distance of the robot, the reflective area can be precisely moved out of the dial, which can provide an accurate basis for the robot pose transformation. Thus, it can avoid cases that the reflection is not completely removed due to insufficient moving distance, or a new reflection is introduced due to excessive moving distance.

In the next section, the proposed method in the above sections will be verified by experiments.

## 5. Experimental Platform and Test Verification

In this section, an experimental platform is described first. Then, some reflective detection examples are presented. The experimental results are compared with those by some typical methods available in the literature. Finally, a robot pose adjusting example is demonstrated.

### 5.1. Experimental Platform and Detection Process

The experimental platform is shown in Figure 16. Its configurations include: NVIDIA Jetson agx Xavier developing board, BTS-1C explosion-proof network cloud platform which contains a camera and an infrared lamp, Autolabor Pro1 robot chassis, PC, display.

Experimental environments of PC are: Intel(R) Core(TM) i5-12400 CPU @ 3.2GHz, 4 GB RAM, Windows 10 64-bit operating system, Python language and Pytorch1.8.1 deep learning framework.

The flow chart of pointer meter detection is shown in Figure 17. Before detection and adjustment of the reflective area of a pointer meter, it is necessary to first collect the image of the pointer meter, and then try to read the pointer meter. If the reading is successful, the task will be completed. If the reading fails, the reflective detection will be carried out to determine whether the reflective area exceeds the set threshold. If the threshold is exceeded, it may be caused by light reflection which affects the reading pointer meter, and the robot pose will be adjusted according to the detection results to complete the reading. If the threshold is not exceeded, it may not be affected by light reflection. The initialization of the reading program will be re-conducted for further detection.

### 5.2. Reflective Area Detection

The improved k-means algorithm is used to detect the reflective area and compared with other detection methods available in the literature. Figure 18 shows the detection results of the proposed detection methods and those in [11,12,13,24]. In this figure, the white parts are identified as the reflective areas of the images detected. It can be seen from Figure 18 that the detection results of the proposed method and those in [13] are good, indicating that the reflective area can be accurately detected, while there are a few false detections by using that in [24]. Additionally, the detection results by using those in [11,12] are poor, including obvious false detections.

In order to further examine the detection speed and accuracy of the proposed method, experiments of 100 times are conducted to detect 592 pixels × 562 pixels (length × width) pointer meter images in Figure 18 collected in four different situations, respectively. The evaluation results of the average detection time and detection accuracy, i.e., the intersection over Union (*IoU*) of the reflective area, are listed in Table 1. Those methods in [11,12,13,24] in the same experimental conditions are also listed in Table 1. The *IoU* here is an evaluation index characterizing the accuracy of the marked reflective area. It is the overlap rate of the marked reflective area (*Ø*) and the subjectively evaluated reflective area (*φ*). It is also the ratio of their intersection to the union. The ideal case is full overlap: *IoU* = 1. Its calculation formula is shown in (17).
(17)$$IoU=\frac{\phi \cap \varphi }{\phi \cup \varphi }$$

Evaluation results in Table 1 show that the proposed method not only has good detection accuracy but also has the least detection time compared with those in [11,12,13,24]. The average detection time is about 0.64 s, which has good real-time performance. There are two main reasons for the fast speed. The first is to perform histogram statistics on the pixel brightness information of the image and divide the pixels into 256 groups. When the k-means are performed according to groups, it will greatly reduce the processing time of the k-means. The second is that an improved k-means initial values method is proposed instead of the traditional random values method, which will shorten the convergence time.

### 5.3. Robot Pose Adjustment

As an example, Figure 19 shows that the robot is adjusted from position A, Figure 19a, to position B, Figure 19b, according to the robot control strategy proposed in this article. The corresponding collected pointer meter images are shown in Figure 20. Figure 20a represents the original pointer meter image with light reflection, and Figure 20b represents the image taken after the robot position adjustment. It can be seen from Figure 20 that the method in this paper and the robot pose control strategy can effectively remove the reflective area from the pointer meter and obtain a high-quality pointer meter image.

## 6. Conclusions

In this paper, to solve the problem of difficult identification and reading caused by the reflection phenomenon in detecting pointer meters by inspection robots, high-quality pointer meter images are obtained by combining machine vision with hardware real-time adjustment. YOLOv5s network and perspective transformation are combined to detect pointer meters based on an improved k-means clustering algorithm. An adaptive detection method for the reflective area of pointer meters and a new robot pose control strategy is proposed. The reflection detection method can avoid the instability of the clustering effect by adaptively determining the optimal number of clusters and the initial clustering center of the k-means algorithm. The pointer meter reflective areas can be removed according to the detection results by using the proposed robot pose control strategy.

Experimental results show that the reflective detection method has good robustness, fast speed, and high accuracy. The robot pose control strategy can effectively adjust the shooting angle to collect high-quality pointer meter images.

The advantages of this proposed method are mainly reflected in several aspects. It can adaptively process pointer meter images under different reflective conditions. There is no need for color segmentation for images. It has low requirements for the continuity of image frames, equipment, and types of images. There is no need to localize reflective areas through interactive prior information. The proposed method can provide a good solution for the pointer meter reading of inspection robots in a reflective environment.

In the practical testing, although the pointer meter recognition rate is more than 95%, there are still very few false detections and missed detections. This is because there are many types of pointer meters, and YOLOv5s has not learned enough features of all pointer meters.

## Figures and Tables

**Figure 1 sensors-23-02562-f001:**
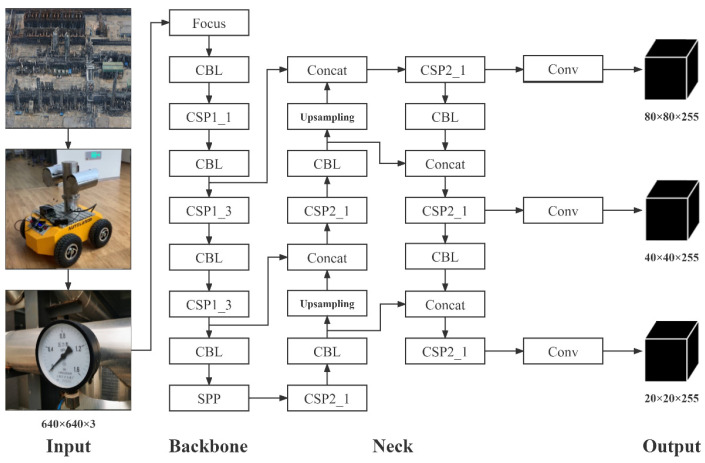
YOLOv5s network structure diagram.

**Figure 2 sensors-23-02562-f002:**
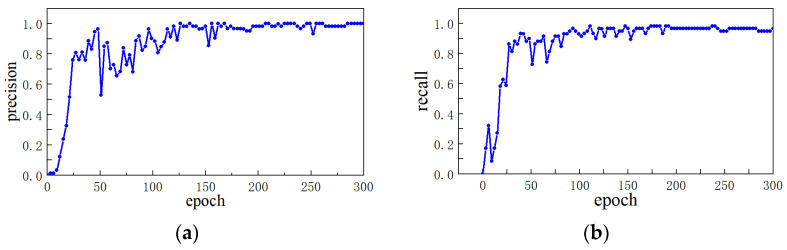
Precision and recall rates: (**a**) precision; (**b**) recall.

**Figure 3 sensors-23-02562-f003:**
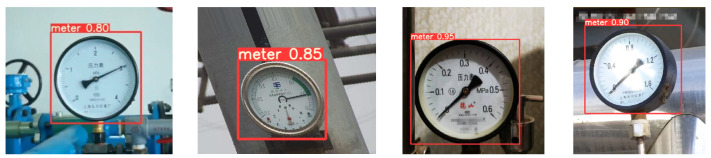
Detection results of pointer meter.

**Figure 4 sensors-23-02562-f004:**
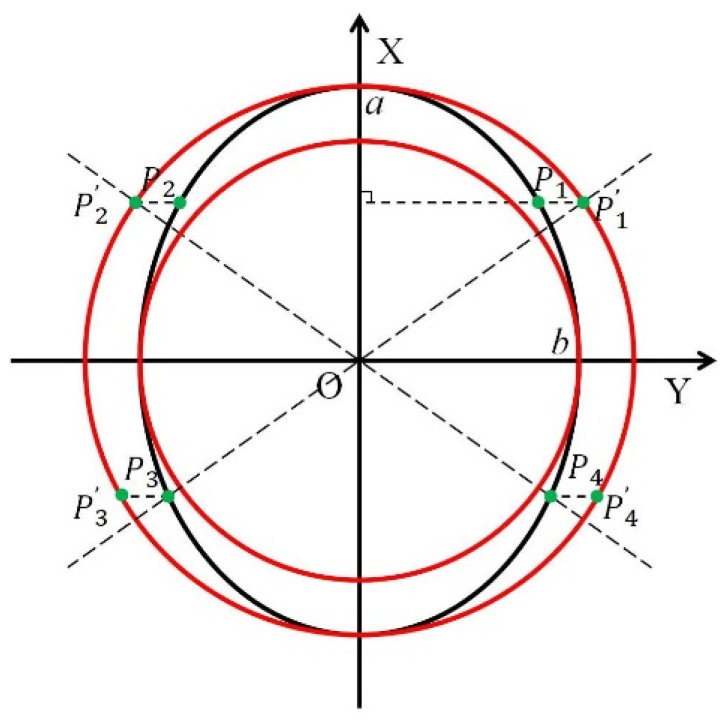
Perspective transformation diagram.

**Figure 5 sensors-23-02562-f005:**
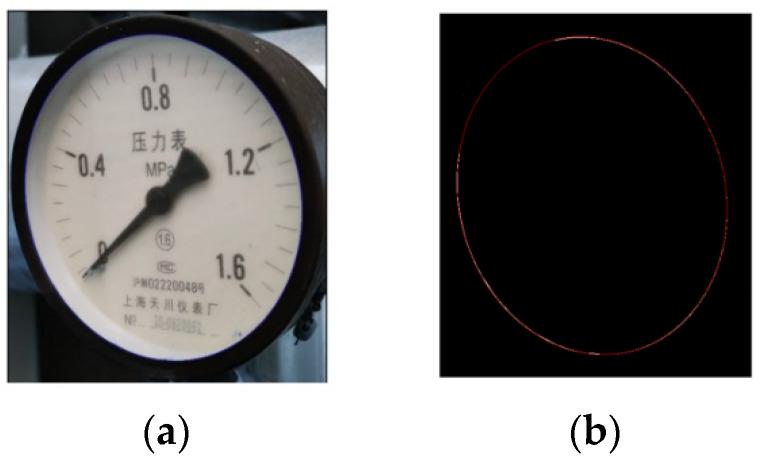
Deep learning and ellipse detection results: (**a**) deep learning detection results; (**b**) ellipse detection results.

**Figure 6 sensors-23-02562-f006:**
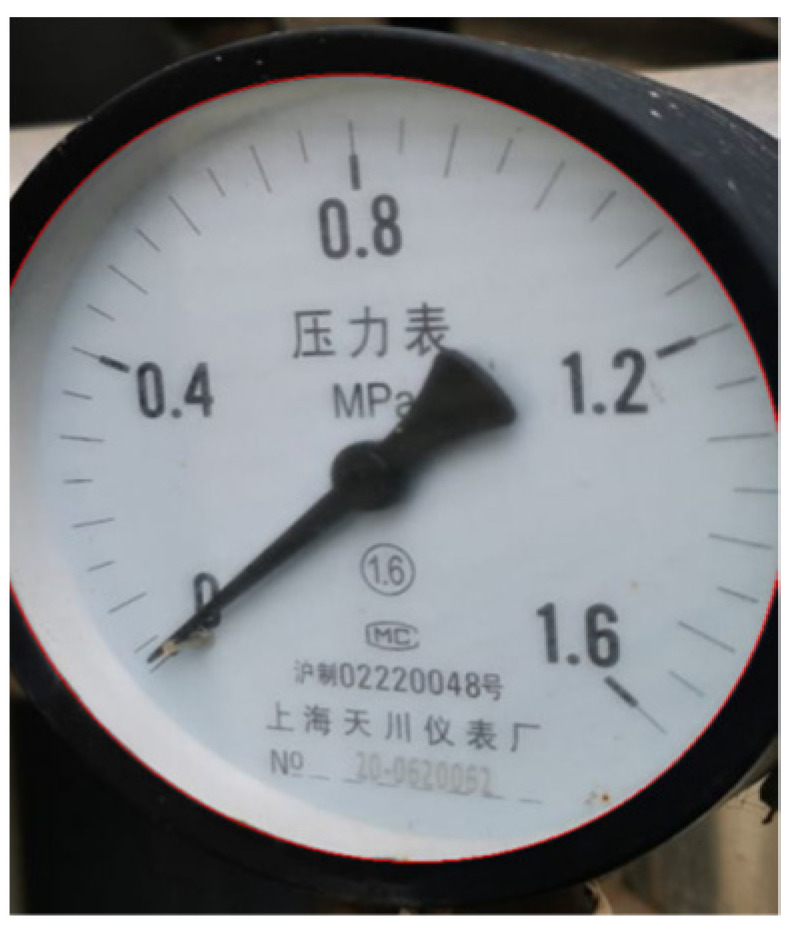
Perspective transformation results.

**Figure 7 sensors-23-02562-f007:**
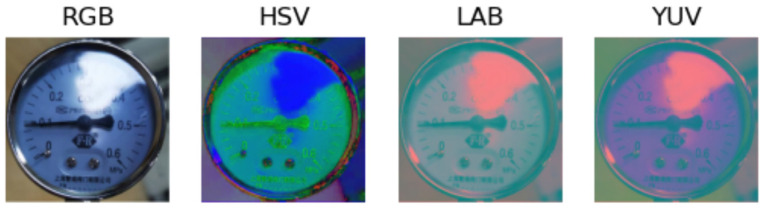
Color model comparison.

**Figure 8 sensors-23-02562-f008:**
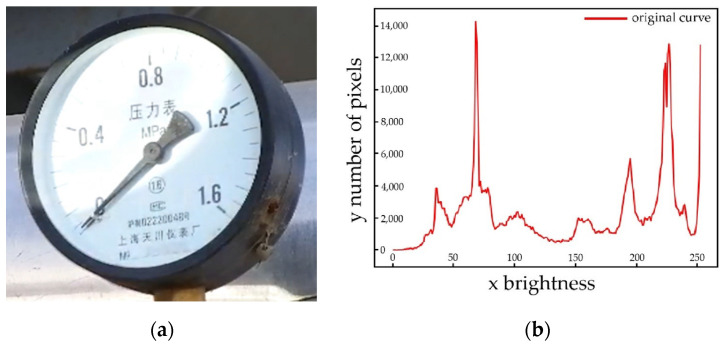
Brightness component histogram; (**a**) original figure; (**b**) histogram.

**Figure 9 sensors-23-02562-f009:**
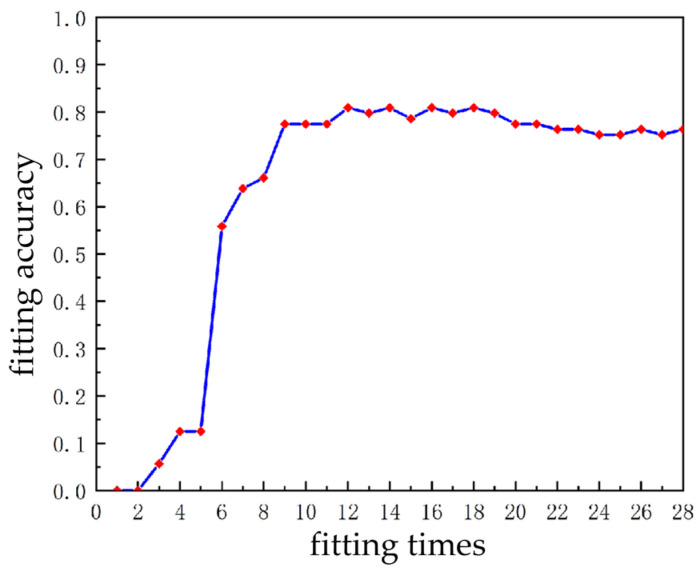
Fitting accuracy vs. fitting times.

**Figure 10 sensors-23-02562-f010:**
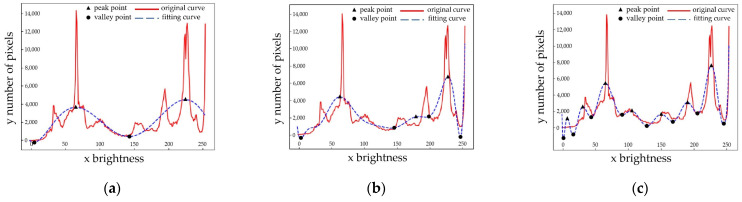
Under-fitting, fitting, and over-fitting curves; (**a**) under-fitting curve; (**b**) fitting curve; (**c**) over-fitting curve.

**Figure 11 sensors-23-02562-f011:**
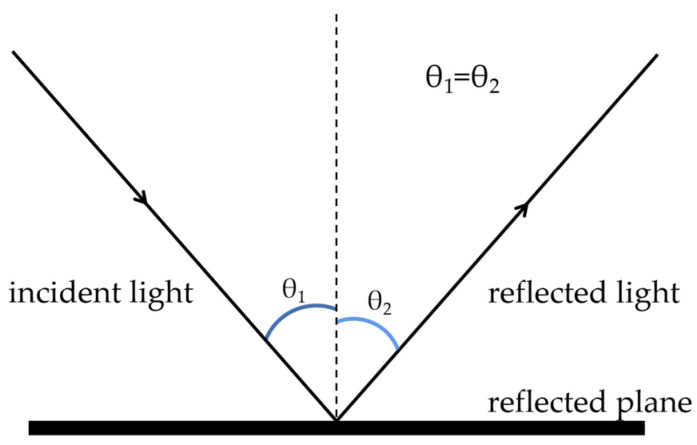
Reflectance law.

**Figure 12 sensors-23-02562-f012:**
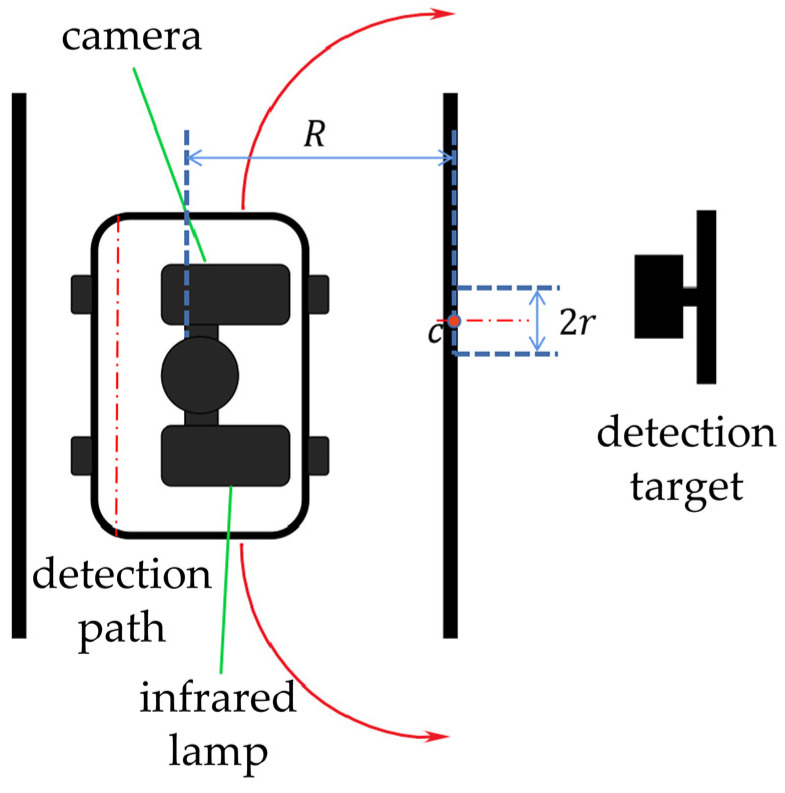
Motion diagram of robot pose adjustment.

**Figure 13 sensors-23-02562-f013:**
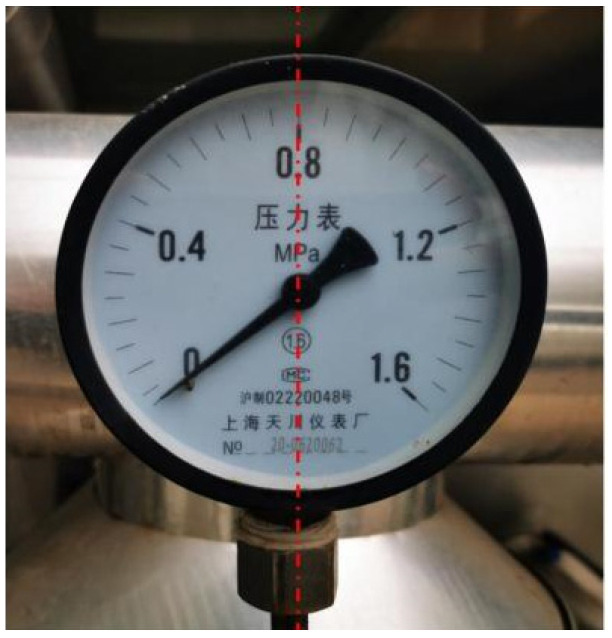
Medial axis diagram of pointer meter.

**Figure 14 sensors-23-02562-f014:**
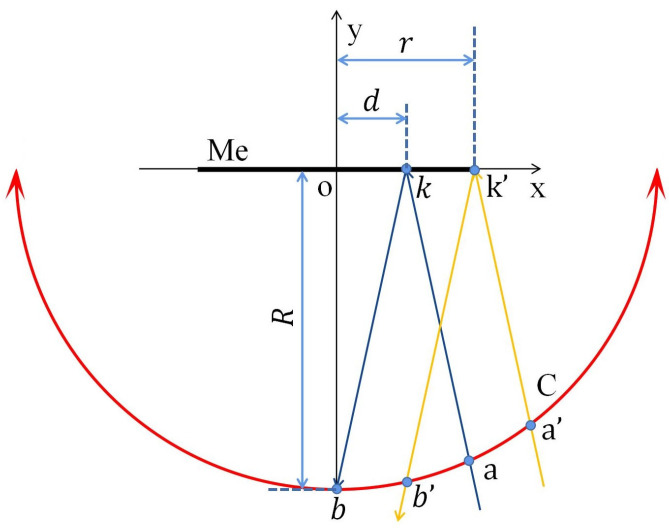
Diagram of robot position relationship.

**Figure 15 sensors-23-02562-f015:**
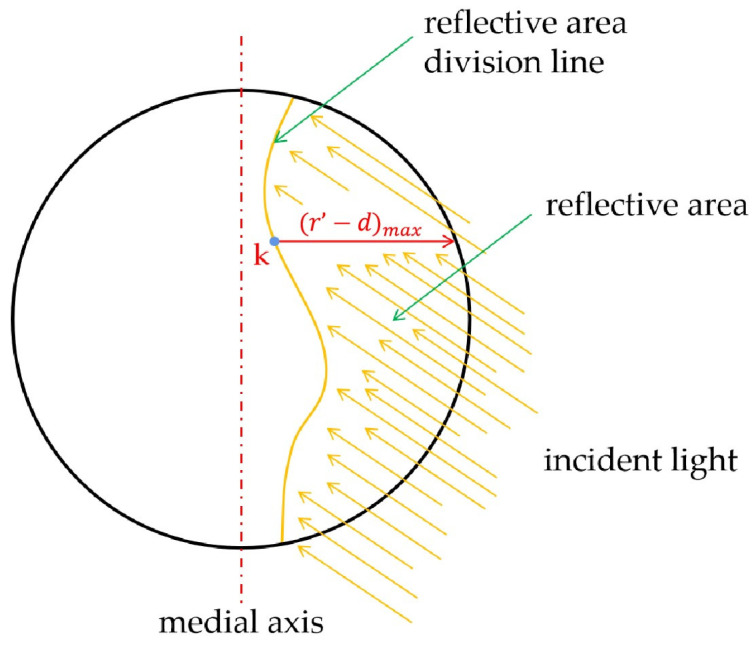
Reflective area diagram.

**Figure 16 sensors-23-02562-f016:**
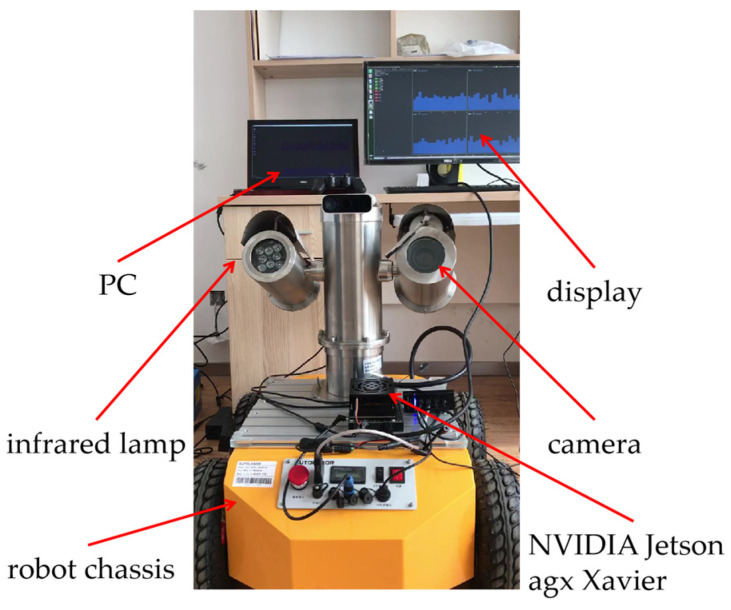
Experimental platform.

**Figure 17 sensors-23-02562-f017:**
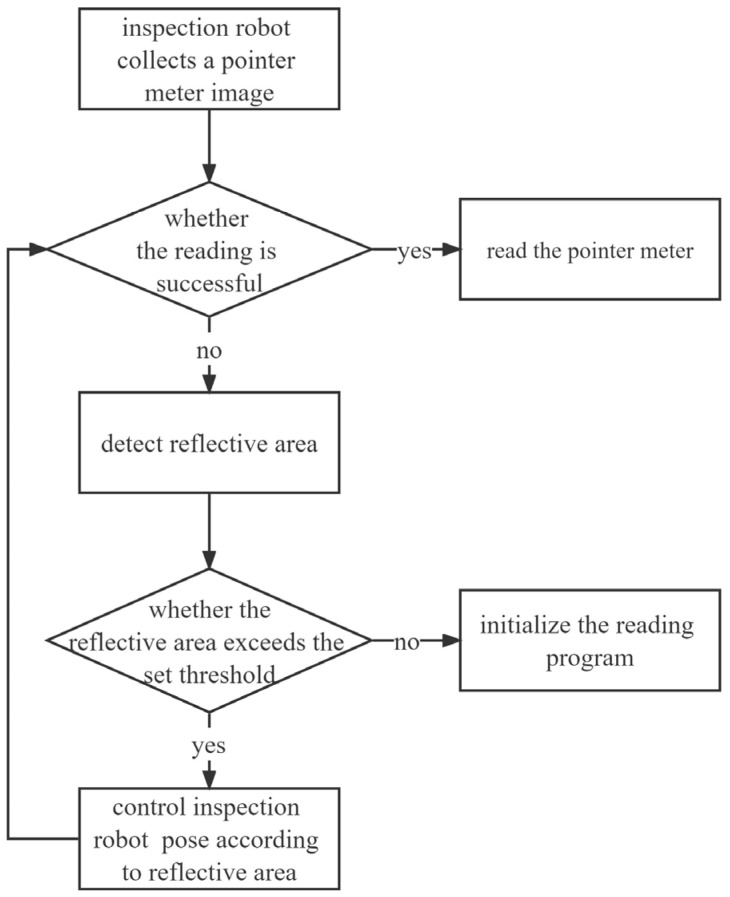
Flow chart of pointer meter detection and recognition.

**Figure 18 sensors-23-02562-f018:**
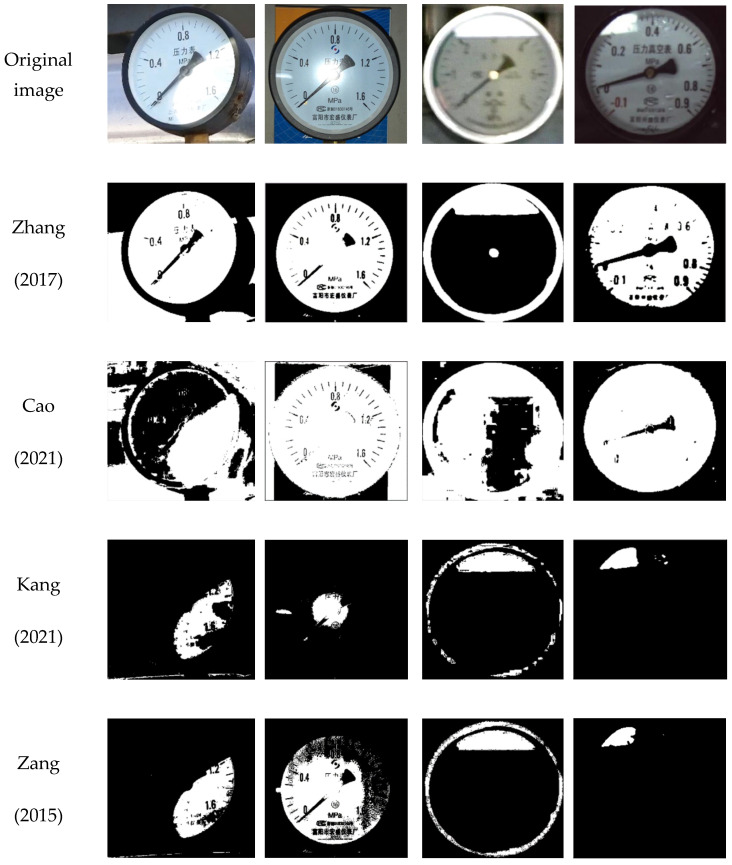
Reflective area detection results [11,12,13,24].

**Figure 19 sensors-23-02562-f019:**
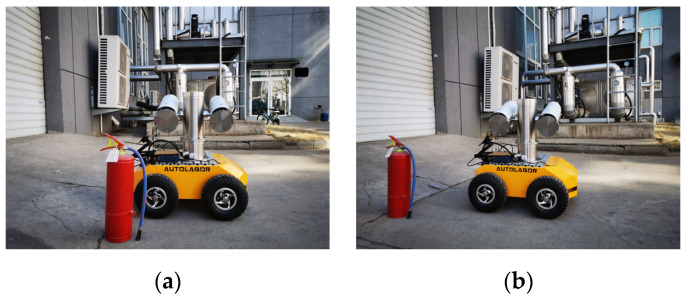
Robot pose adjustment experiment; (**a**) position A; (**b**) position B.

**Figure 20 sensors-23-02562-f020:**
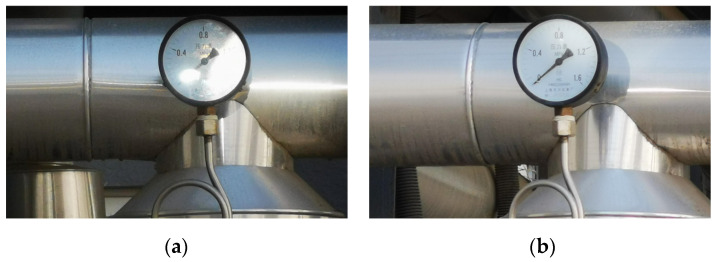
Comparison of light reflection adjustment; (**a**) image without removing light reflection; (**b**) image after removing light reflection.

**Table 1 sensors-23-02562-t001:** Comparison of evaluation results.

Method	Zhang [11] (2017)	Cao [12] (2021)	Kang [13] (2021)	Zang [24] (2015)	Ours
detection time/s	1.0477	3.1545	2.8415	1.3349	0.6392
detection accuracy	0.143	0.611	0.833	0.727	0.809

## Data Availability

The data presented in this study are available on request from the corresponding author.

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
