# Peer review of "Adaptive Reflection Detection and Control Strategy of Pointer Meters Based on YOLOv5s"

_sensors, 2023, doi:10.3390/s23052562_

Round 1

Reviewer 1 Report

The authors proposed an adaptive detection method of pointer meter reflective areas and a robot pose control strategy based on deep learning and an improved k-means clustering algorithm. The following comments need to be considered. 

1) The novelty and contributions are not clear enough in the abstract. 

2) In the abstract, write the full names of YOLOV5S and YUV. 

3) In the introduction, write the main contributions in bullets and state advantages and disadvantages of the proposed method. 

4) More literature review is required with paying attention to the recent publications 

5) Fix typos, example see line 112. 

6) Table 1 shows the proposed algorithm outperforms the existing ones in terms of efficiency and processing time. Explain in details why and what cost for this great performance. 

7) Define equations parameters. Normally I do not find their definitions. 

8) Provide more inside on figure 10.  

Reviewer 2 Report

Major revision

In this manuscript, an adaptive instrument reflection detection method is proposed to solve the problem that the inspection robot detects the protective glass of pointer instruments in complex environment, and an optimized robot attitude strategy is proposed accordingly. This method is helpful to improve the automation of inspection work in industrial environment to some extent and has certain practicability. However, there are still some problems and mistakes to be improved in this manuscript, as follows:

1. The pointer instruments used in petrochemical industry are usually protected by crystal glass, and the reflected light generated by crystal glass can be easily weakened or even eliminated by multilayer polarization filters. The author of this manuscript also mentioned relevant research results in line 57. However, the proposed method does not take this into consideration. It would be beneficial to compare the proposed method with this existing solution.

2. While the algorithm presented in the paper has some practical advantages, the research significance is not entirely clear. Maintaining an inspection robot can be more difficult and less reliable than manual inspection. Instead of pursuing an all-terrain inspection robot, it might be more practical to focus on designing a low-cost monitoring system. The authors may consider reorienting the research focus towards a different scenario.

3. The authors may add more state-of-art articles for the integrity of the manuscript (Rachis detection and three-dimensional localization of cut off point for vision-based banana robot. Computers and Electronics in Agriculture 2022. A Study on Long–Close Distance Coordination Control Strategy for Litchi Picking. Agronomy 2022).

4. The abstract presents a clear overview of the document, but it would be helpful to include numerical results from the research in the section on results.

5. Briefly mentioning recent developments in the YOLO family of algorithms (such as YOLOv6, v7, and v8) would provide additional context for the study.

6. In Chapter 3.1, RGB, HSV, and YUV are not referred to as "color spaces".

7. Furthermore, the brightness channel of HSV is not independent of the hue and saturation channels, which are inaccurately described in the chapter. The authors should consult relevant literature in this field and revise the chapter accordingly.

8. Vision technology integrated with deep learning is emerging these years in various engineering fields. For civil applications, references keywords such as integrated generative adversarial networks and improved VGG model, references keywords such as backbone double-scale features for improved detection automation, should be mentioned in the introduction of method part.

9. Figure 7 may have misused a picture or not accurately understood the color representation model. It appears that the original RGB three-channel data was used as the three-channel data for HSV and other representations without conversion. To study the impact of reflection on patrol inspection, it would be beneficial to compare dial inspection results of different methods, as the author has not discussed how the data will be read manually after the robot captures it.

10. In Figure 19, the difference before and after robot posture adjustment is not clearly visible. The authors should consider using a picture with a more noticeable difference.

11. In Figure 20, the perspective relationship in the pictures appears to have "almost" not changed, and it is unclear how reflection was addressed. If possible, the authors should consider uploading relevant videos to clarify.

Round 2

Reviewer 1 Report

thank you 

Reviewer 2 Report

I recommend the publication of this manuscript since the authors have successfully addressed all the comments.